# Removal of Aqueous Cu²⁺ by Amorphous Calcium Carbonate: Efficiency and Mechanism

**Zisheng Liao** [1,2,3]**, Shijun Wu** [1,2,*] **, Hanxiao Zhang** [1,2,3] **and Fanrong Chen** [1,2]

1   CAS Key Laboratory of Mineralogy and Metallogeny & Guangdong Provincial Key Laboratory of Mineral Physics and Materials, Guangzhou Institute of Geochemistry, Chinese Academy of Sciences, 511 Kehua Street, Guangzhou 510640, China; liaozisheng19@mails.ucas.ac.cn (Z.L.); zhanghanxiao18@mails.ucas.ac.cn (H.Z.); frchen@gig.ac.cn (F.C.)
2   CAS Center for Excellence in Deep Earth Science, 511 Kehua Street, Guangzhou 510640, China
3   University of Chinese Academy of Sciences, 19 Yuquan Road, Beijing 100049, China
*   Correspondence: wus@gig.ac.cn; Tel.: +86-20-8529-0143

**Abstract:** Crystalline calcium carbonate ($CaCO_3$, such as calcite) could scavenge aqueous metals via adsorption and coprecipitation. As a precursor to crystalline $CaCO_3$, amorphous calcium carbonate (ACC) is poorly understood on metals removal. Herein, we synthesized silica-stabilized ACC and investigated its $Cu^{2+}$ removal efficiency and mechanism. The results showed that the $Cu^{2+}$ removal efficiency by ACC is controlled by the initial solution pH, initial $Cu^{2+}$ concentration, contacting time, and ACC dosage. The maximum $Cu^{2+}$ removal capacity was 543.4 mg/g at an ACC dosage of 1 g/L, an initial pH of 5.0, an initial $Cu^{2+}$ concentration of 1000 mg/L, and an equilibrium time of 20 h. X-ray powder diffraction (XRD) and scanning electron microscope with an energy dispersive spectrometer (SEM-EDS) revealed that $Cu^{2+}$ precipitated as paratacamite ($Cu_2(OH)_3Cl$, space group: $R\bar{3}$) at an ACC dosage of 1 g/L, whereas botallackite ($Cu_2(OH)_3Cl$, space group: $P2_1/m$) was the Cu-bearing product for crystalline calcite using the same dosage as ACC. However, $Cu^{2+}$ preferred to incorporate into calcite, which is transformed from ACC at high ACC loading (such as 4 g/L). Our results demonstrated that the crystallinity and dosage of $CaCO_3$ could control the $Cu^{2+}$ removal mechanism.

**Keywords:** amorphous materials; calcium carbonate; copper removal; wastewater

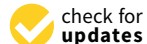



## 1. Introduction

Heavy metals (HMs) pollution in surface water is a global environmental problem. There were increasing trends for Cd, Cr, Cu, Ni, Mn, and Fe in global rivers and lakes from 1970 to 2017, whereas the mean dissolved concentrations of most HMs were the highest in Asia [1]. For example, the average concentration of HMs in the Upper Ganga River in India was high, mostly exceeding the limits prescribed for surface water by the Bureau of Indian Standard (BIS) and the World Health Organization (WHO) [2]. Similarly, the concentrations of Cr, As, Cd, and Pb in the Korotoa River in Bengal were higher than the safe recommended values, adversely affecting this riverine ecosystem [3].

As an essential element, Cu is one of the indispensable trace elements for the human body. Trace copper is conducive to accelerating growth and normal physiological function, whereas excess threatens living beings [4]. Cu exists in the forms of $Cu^+$, $Cu^{2+}$, and $Cu^{3+}$ ions, among which $Cu^{2+}$ is the most stable and widespread in the environment [5,6]. $Cu^{2+}$ is usually found at high concentrations in industrial wastewater, such as metal finishing, electroplating, plastics, and etching. Currently, there are many technologies to treat copper-contaminated wastewater, including adsorption, chemical precipitation, membrane filtration, and electrochemical methods [7,8]. Among them, adsorption and precipitation were the more convenient and economical ways to treat HMs wastewater than others.

Calcium carbonate ($CaCO_3$) has attracted lots of attention in copper removal because calcium carbonate minerals can combine with HMs through adsorption and coprecipita-

tion [9–16]. Limestone was reported with similar copper-removal efficiency to activated carbon, and the efficiency of HMs removal could be better for natural limestone, which contained higher amounts of silica and iron/aluminum oxides [10–12]. Cu can be selectively separated from other metals (Ni, Mn, Zn, and Cd) by using mechanically activated calcium carbonate, with posnjakite ($Cu_4(SO_4)(OH)_6 \cdot H_2O$) as the main product [13,14]. Furthermore, different anions presented together with Cu could result in varied precipitates, such as posnjakite ($Cu_4(SO_4)(OH)_6 \cdot H_2O$) for $SO_4^{2-}$, gerhardtite ($Cu_2(NO_3)(OH)_3$) for $NO_3^-$, atacamite ($Cu_2(OH)_3Cl$, space group: *Pnma*) and botallackite ($Cu_2(OH)_3Cl$, space group: $P2_1/m$) for $Cl^-$ [16]. Previous studies also demonstrated that $Cu^{2+}$ could be integrated into the calcite lattice during the transformation of vaterite into calcite [17–20].

CaCO₃ exhibits five crystalline polymorphs: ikaite ($CaCO_3 \cdot 6H_2O$), monohydrocalcite ($CaCO_3 \cdot H_2O$), calcite ($CaCO_3$, space group: $R\bar{3}c$), aragonite ($CaCO_3$, space group: *Pmcn*), and vaterite ($CaCO_3$, space group: $P6_3/mmc$), and one amorphous phase: amorphous calcium carbonate ($CaCO_3 \cdot nH_2O$, ACC) [21]. As an unstable phase, ACC could transform to crystalline phases by the solid-state transformation pathway or dissolution–recrystallization [22]. It is well known that the precipitation of CaCO₃ could scavenge metals in the solution [23]. Meanwhile, ACC is highly enriched in trace elements relative to crystalline carbonates [24]. When impurity-bearing ACC crystallizes to calcite, enhanced uptake of impurity ions can be achieved [25,26]. However, all CaCO₃ mentioned above used to remove Cu were crystallized phase. Only one poly(acrylic acid) stabilized ACC was used to remove aqueous $Cd^{2+}$, $Pb^{2+}$, $Cr^{3+}$, $Fe^{3+}$, and $Ni^{2+}$, but the mechanism is not well understood [27].

Herein, we prepared silica-stabilized ACC and investigated its $Cu^{2+}$ removal ability compared with crystallized calcite. X-ray diffraction (XRD) and a scanning electron microscope with an energy dispersive spectrometer (SEM-EDS) were used to characterize the products. The $Cu^{2+}$ removal mechanism by ACC was proposed to be precipitation and incorporation. Our results provided further understanding of the HMs removal mechanism by ACC and the potential application of ACC as an HMs treatment material.

## 2. Materials and Methods

### 2.1. Synthesis of ACC

All chemical reagents were analytical grade except hydrochloric acid and nitric acid of superior pure grade. The synthesis method of silica-stabilized ACC was modified after Kellermeier et al. [28]. Two ACC, i.e., AmCalA and AmCalB, with molar ratios of Si/Ca = 1 and 2, were prepared and used in this study. Briefly, aqueous solutions of 1 M CaCl₂, 1 M NaCO₃, and 1 M Na₂SiO₃ (adjusted to pH 12.4 with 1 M NaOH) were prepared in deionized water separately and then stored at −4 °C for at least 2 h. After that, a specific volume of 1 M Na₂SiO₃ was mixed with 1 M CaCl₂ according to the corresponding molar ratios of Si/Ca, and then 1 M NaCO₃ in the same volume as 1 M CaCl₂ was added. The reaction solution was kept stirring for 5 min in an ice-water bath. After centrifugation, the precipitates were washed with deionized water (~0 °C) one time and freeze-dried in a vacuum for further characterization.

### 2.2. $Cu^{2+}$ Removal by ACC

ACC passed through a 0.25 mm sieve was mixed with copper chloride solution and stirred at a speed of 150 r/min at room temperature (~25 °C) for a specific time. At the end of the reaction, the solution pH was recorded. After centrifugation, the supernatant was filtered with a 0.45 μm filter and used for $Cu^{2+}$ concentration analysis. The rest of the precipitate was freeze-dried in a vacuum and stored for further characterizations.

#### 2.2.1. Effect of Initial pH

The different initial pH values (from 1.0 to 8.0) of 500 mg/L $Cu^{2+}$ solutions were adjusted using 1 mol/L HCl and 1 mol/L NaOH. AmCalA was stirred with $Cu^{2+}$ solution in a solid/liquid ratio of 1 g/L for 24 h.

### 2.2.2. Effect of Contact Time

At an initial pH value of 5, 500 mL $Cu^{2+}$ solution (500 mg/L) was reacted with AmCalA (1 g/L) under magnetic stirring. Ten mL liquid was sampled and centrifuged at a specific time (from 5 min to 28 h).

### 2.2.3. Effect of Initial $Cu^{2+}$ Concentration

$Cu^{2+}$ solutions with different initial concentrations (200, 300, 400, 500, 600, 700, 800, 900, and 1000 mg/L) were prepared by $CuCl_2$, whose pH was later adjusted to 5.0. One g/L AmCalA was used for the experiment with a reaction duration of 20 h.

### 2.2.4. Effect of Material Dosage

AmCalA with different dosages (from 0.4 to 2.0 g/L) was reacted with a 500 mg/L $Cu^{2+}$ solution with an initial pH of 5.0 for 20 h.

### 2.2.5. Comparison of $Cu^{2+}$ Removal Ability by Different Calcium Carbonate

Two crystallized $CaCO_3$ were used to compare the $Cu^{2+}$ removal ability with AmCalA and AmCalB. The self-synthesized calcite (CryCal) was prepared using the same method as AmCal without using $Na_2SiO_3$, and one commercial $CaCO_3$ (ComCal) was purchased from Shanghai Macklin Biochemical Co., Ltd. Each $CaCO_3$ was reacted with a 700 mg/L $Cu^{2+}$ solution with an initial pH of 5.0, a dosage range of 1.0~4.0 g/L, and contact time of 20 h. The obtained solid products were named calcite_X, where X indicates the material dosage. For example, AmCalA_1 was the solid product of AmCalA after $Cu^{2+}$ removal, whose initial dosage was 1 g/L.

### 2.3. Characterization

The X-ray powder diffraction (XRD) patterns were obtained using a Brucker D8 AdvanceX diffractometer (BRUKER D8 ADVANCE, Bruker Corporation, Billerica, MA, USA) with a Cu anode (40 kV and 40 mA) between 10° and 80° (2θ) with a step speed of 5° (2θ) per minute. The phases were identified based on the International Centre for Diffraction Data (ICCD) Database. Semiquantitative analysis was carried out using RIR values by Jade 6.5.

A field emission scanning electron microscope (SEM: ZEISS Ultra 55, Carl Zeiss, Oberkochen, Germany) was used to observe the morphology of the precipitates with an accelerating voltage of 1.5 kV. Meanwhile, the attached energy dispersive X-ray spectroscopy (EDS: Oxford X-Max 50, Oxford Instruments, Oxford, England) analyzed the chemical composition.

The concentrations of Cu, Ca, and Si in the filtrate solution were determined by an inductively coupled plasma atomic emission spectrometer (ICP-AES: Varian Vista-Pro, Varian, Palo Alto, CA, USA). A pH meter was used to measure the solution pH (Seven Excellence, Mettler Toledo, Shanghai, China).

### 2.4. Data Processing

The $Cu^{2+}$ removal efficiency (R ∈ [0,100], %) and removal capacity ($q_e$, mg/g) by particular $CaCO_3$ were calculated using Equations (1) and (2).

$$R = \frac{C_0}{C_e} \times 100\% \tag{1}$$

$$q_e = (C_0 - C_e) \times \frac{V}{m} \tag{2}$$

where $C_0$ and $C_e$ (mg/L) are the initial and equilibrium concentrations of $Cu^{2+}$, respectively, V is the volume of the solution (L), and m is the mass of used $CaCO_3$ (g).

## 3. Results and Discussion

### 3.1. Characterization of CaCO₃

As shown in Figure 1a, the XRD patterns of self-synthesized calcite (CryCal) and commercial CaCO₃ (ComCal) possessed typical reflections as crystallized calcite (CaCO₃). The strong reflections at 31.7°, 45.5°, and 56.5° (2θ) in AmCalA and AmCalB could be assigned to the (200), (220), and (222) plane of halite (NaCl). Theoretically, two-mole halite will be produced along with a one-mole CaCO₃ precipitate, as shown in Equation (3). Though NaCl is soluble, we only washed the precipitate one time to avert ACC crystallization. The residual NaCl in the solution likely crystallized to halite during vacuum drying. Furthermore, AmCalA showed one weak reflection at 29.4° (2θ), corresponding to calcite's (104) plane. Because other calcite reflections were absent, AmCalA was regarded as weak crystalline ACC mixed with halite. Furthermore, no other reflections besides halite were observed in the XRD pattern of AmCalB. A broad weak peak at 28°~31°(2θ) suggests the amorphous characterization.

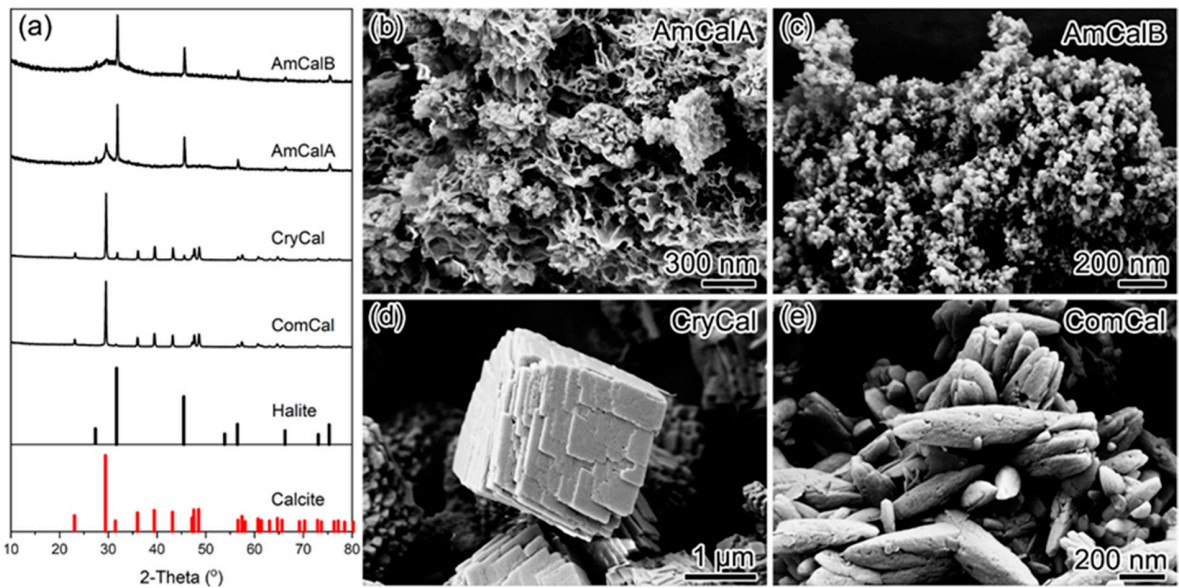

**Figure 1.** The powder XRD patterns (**a**) and the SEM images of varied CaCO₃, (**b**) AmCalA, (**c**) AmCalB, (**d**) CryCal, and (**e**) ComCal.

SEM images showed that AmCalA adopts an aggregated nano-fragmented morphology (Figure 1b), whereas AmCalB was an aggregated nanoparticle (Figure 1c), which supports the presence of amorphous material [28]. CryCal showed a typical rhombohedral habit of calcite with a growth hillock on the surface (Figure 1d). Contrarily, ComCal was aggregated crystals with a spindle shape (Figure 1e).

$$CaCl_2 + Na_2CO_3 \rightarrow CaCO_3 \downarrow + 2NaCl \tag{3}$$

### 3.2. Removal of Cu²⁺ by Amorphous Calcium Carbonate

#### 3.2.1. Effect of Initial pH

As a typical transition element metal, the stability of $Cu^{2+}$ in a solution is controlled by the solution pH. We first checked $Cu^{2+}$ concentration in a stock solution with varied pH and found that $Cu^{2+}$ started to precipitate at pH 6.0 and precipitately completely at pH 7.0 (Supplementary Materials Figure S1). Based on the simulation using Visual MINTEQ, the saturation index of atacamite is 0.997 in a 500 mg/L $Cu^{2+}$ (in the form of $CuCl_2$) solution at pH 5.0. However, we did not observe any precipitates at this condition, suggesting this solution is oversaturated. After reacting with AmCalA, the removal efficiencies (R) increased quickly from 1.1% to 28.7% and 93.6%, with the initial pH of 1.0, 2.0, and 3.0

(Figure 2a). Then R increased to 93.6% ($pH_{initial}$ = 4.0), 96.5% ($pH_{initial}$ = 5.0), and 99.2% ($pH_{initial}$ = 6.0). Meanwhile, the final solution pH could be divided into three groups. Group 1 includes sole data, i.e., 1.1, which increased slightly compared to the initial pH 1.0. Group 2 includes four data (4.8, 5.6, 5.8, and 6.1) with initial pH in the range between 2.0 and 5.0. When the initial pH was 6.0, the final pH changed to 10.6, Group 3. The increase of solution pH was ascribed to the consumption of $H^+$ that reacted with $CaCO_3$, and hydrolysis of the carbonate ion, as described by Equations (4)–(6). This result indicated that the solution pH controls the $Cu^{2+}$ removal efficiencies, supported by the positive relationship analysis with $R^2$ values of 0.76 ($pH_{initial}$) and 0.60 ($pH_{final}$). After that, initial pH of 5.0 was selected for the later experiments.

$$2H^+ + CaCO_3 \rightarrow Ca^{2+} + H_2O + CO_2 \uparrow \qquad (4)$$

$$CaCO_3 \rightarrow Ca^{2+} + CO_3^{2-} \qquad (5)$$

$$CO_3^{2-} + H_2O \rightarrow HCO_3^- + OH^- \qquad (6)$$

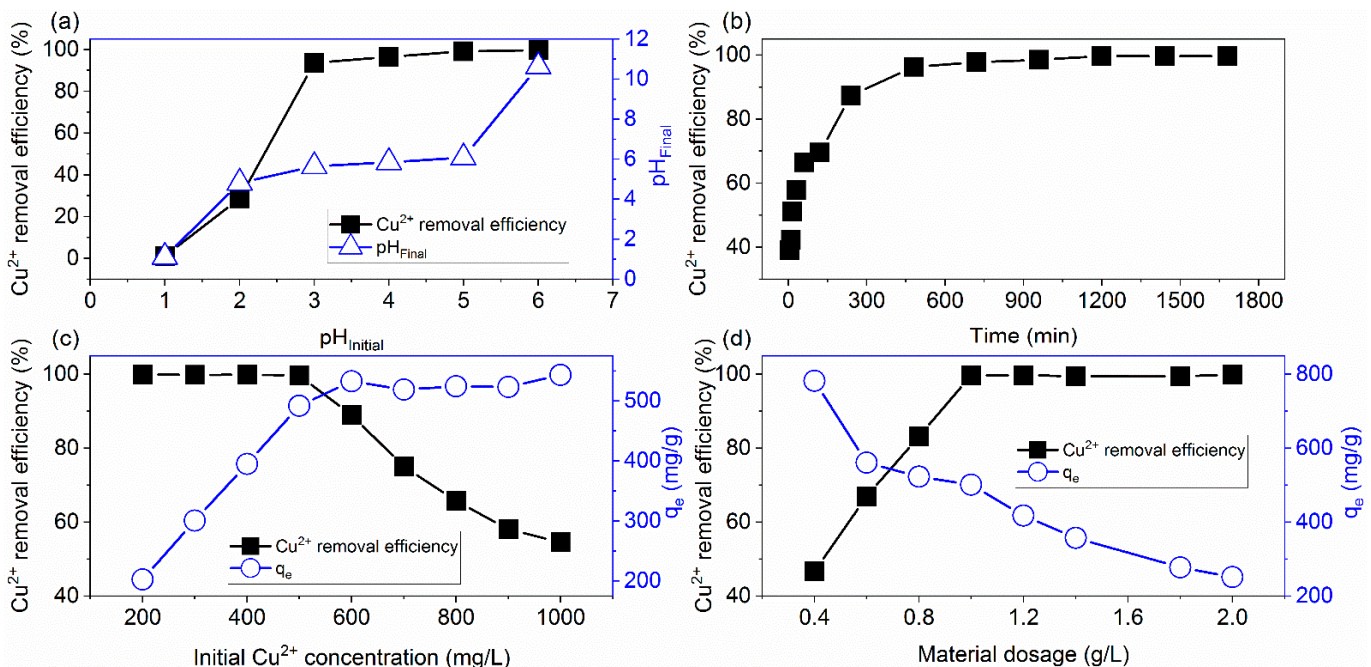

**Figure 2.** (**a**) The relationship between $Cu^{2+}$ removal, final solution pH, and the initial pH. (**b**) The effect of reaction time on $Cu^{2+}$ removal. (**c**) The relationship between $Cu^{2+}$ removal and its initial concentration. (**d**) The evolution of $Cu^{2+}$ removal ability with the dosage of AmCalA.

### 3.2.2. Effect of Contact Time

After contacting for five minutes, 39.1% $Cu^{2+}$ has been removed (Figure 2b), suggesting fast reaction dynamics. Then, the removal efficiency increased to 69.5%, 96.3%, and 99.7% at 2 h, 8 h, and 20 h, respectively. Afterward, the removal efficiency kept stable, implying the equilibrium time was 20 h. Therefore, a contact time of 20 h was used in the other experiments.

### 3.2.3. Effect of Initial $Cu^{2+}$ Concentration

Figure 1c shows that nearly 100% $Cu^{2+}$ had been removed with the initial $Cu^{2+}$ concentration between 200 and 500 mg/L. Meanwhile, the removal capacity increased from 202.6 to 492.1 mg/g. When the initial $Cu^{2+}$ concentration was 600 mg/L, the R decreased to 89.0%, whereas $q_e$ increased to 532.7 mg/g. Furthermore, R decreased until 54.6% at an initial $Cu^{2+}$ concentration of 1000 mg/g, with $q_e$ of 543.4 mg/g. Thus, 500 mg/L was regarded as the maximum $Cu^{2+}$ concentration at a dosage of 1 g/L.

### 3.2.4. Effect of Material Dosage

As shown in Figure 1d, $Cu^{2+}$ removal efficiency gradually increased from 44.7% to 99.7% at the AmCalA dosages ranging from 0.4 to 1.0 g/L. It increased slightly to 99.8% with the dosage increase and reached 100% when the dosages were 1.2 and 4.0 g/L, respectively. On the other hand, the maximum removal capacity was 782.4 mg/g at a dosage of 0.4 g/L. Then, R decreased step by step with the dosage increase until it reached the minimum value of 251.4 mg/g at a dosage of 2.0 g/L. It is evident that the removal efficiency and removal capacity were positively ($R^2$ = 0.997) and negatively ($R^2$ = 0.786) related to the dosage, respectively, especially in the range between 0.4 and 1.0 g/L.

### 3.3. Comparison of $Cu^{2+}$ Removal Ability by Different Calcium Carbonate

### 3.3.1. $Cu^{2+}$ Removal Capacity of Different Calcium Carbonate

Surprisingly, the $Cu^{2+}$ removal efficiency by ComCal and CryCal were 97.9% and 99.2% at a dosage of 1 g/L, and these data were 1.3 times that by AmCalA (78.0%) and AmCalB (75.9%) (Figure 3a). Accordingly, the removal capacities were 698.8 (ComCal), 708.1 (CryCal), 556.9 (AmCalA), and 542.0 mg/g (AmCalB), respectively (Figure 3b). When the dosage was 2 g/L, all removal efficiencies were close to 100% except AmCalB (98.2%), though the removal capacities decreased to 350.7~356.8 mg/g. As hydrous-bearing $CaCO_3$, AmCalA and AmCalB contained both water and impurity silicon in their structure (though we did not qualify it). Therefore, their valid $CaCO_3$ content is lower than the crystalline material, resulting in less $Cu^{2+}$ removal capacity at the dosage of 1 g/L.

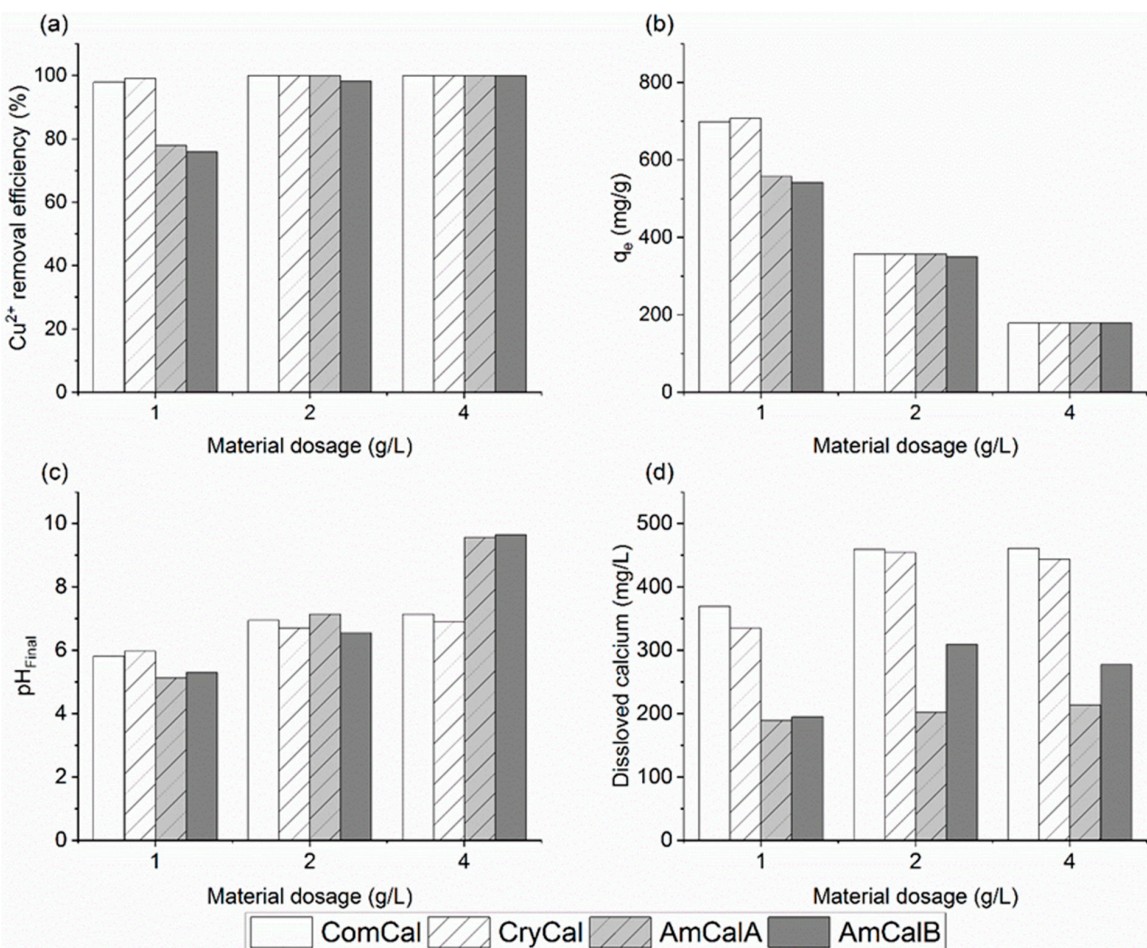

**Figure 3.** Comparison of $Cu^{2+}$ removal ability by various calcium carbonates: (**a**) removal efficiency; (**b**) removal capacity; (**c**) final solution pH, and (**d**) dissolved calcium.

After the reaction, 1 g/L CryCal and ComCal improved the solution pH to 5.8 and 6.0, slightly higher than AmCalA and AmCalB (5.1 and 5.3) (Figure 3c). As discussed in Section 3.2.1, before the removal efficiency reached 100%, it was positively related to the final solution pH. With the dosage increase to 2 g/L and 4 g/L, the final solution pH changed to 6.5~7.1 and 6.9~9.5, independent of the removal efficiencies. Meanwhile, the crystalline $CaCO_3$ always released much more calcium into the solution than the amorphous phases (Figure 3d). For example, the dissolved calcium were 369.5, 334.4, 189.1, and 195.0 mg/L for CryCal, ComCal, AmCalA, and AmCalB at a dosage of 1 g/L, respectively. Besides calcium, ACC released partial silicon into the solution, as shown in Supplementary Materials Figure S2.

### 3.3.2. XRD Characterization of Products after $Cu^{2+}$ Removal

Figure 4 shows the powder XRD patterns of products after copper removal. Besides calcite ($CaCO_3$), both botallackite ($Cu_2(OH)_3Cl$, space group: $P2_1/m$) and paratacamite ($Cu_2(OH)_3Cl$, space group: $R\overline{3}$) were observed. Botallackite could be identified according to the strong reflections at 15.5°, 34.8°, and 37.4° (2θ), which correspond to the (100), (−102), and (121) plane of botallackite. Similarly, the three strong reflections at 16.3°, 32.4°, and 39.8° (2θ) indicated the presence of paratacamite. No halite (NaCl) occurred in the products anymore.

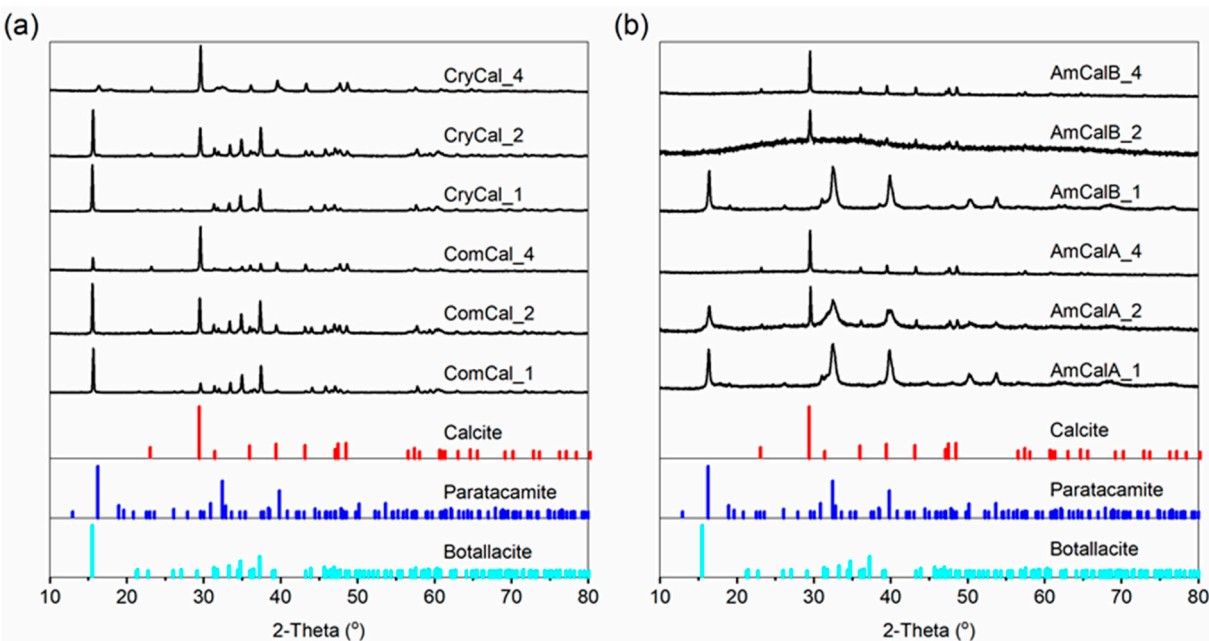

**Figure 4.** Powder XRD patterns of the products after $Cu^{2+}$ removal with different material dosage. (**a**) CryCal and ComCal, (**b**) AmCalA and AmCalB.

Semiquantitative results showed that the products of crystalline $CaCO_3$ were dominated by botallackite (83% and 100%), whereas ACC was dominated by paratacamite (100%) at a dosage of 1 g/L (Figure 5). With the dosage increase to 2 g/L, the proportion of botallackite decreased to 44% and 62% for ComCal and CryCal, whereas paratacamite decreased to 37% for AmCalA. The second mineral phase in the products was calcite, which was unreacted crystalline $CaCO_3$ or transformed crystalline products of ACC. Noteworthy, calcite was the sole mineral phase of AmCalB's product. When the material dosage was 4 g/L, AmCalA also produced only calcite, whereas the ComCal and CryCal produced calcite dominate, plus ~10% botallackite or paratacamite. It is worth noting that the above semiquantitative results of XRD are not accurate and are only a general reference because materials containing trace impurities will alter peak intensity.

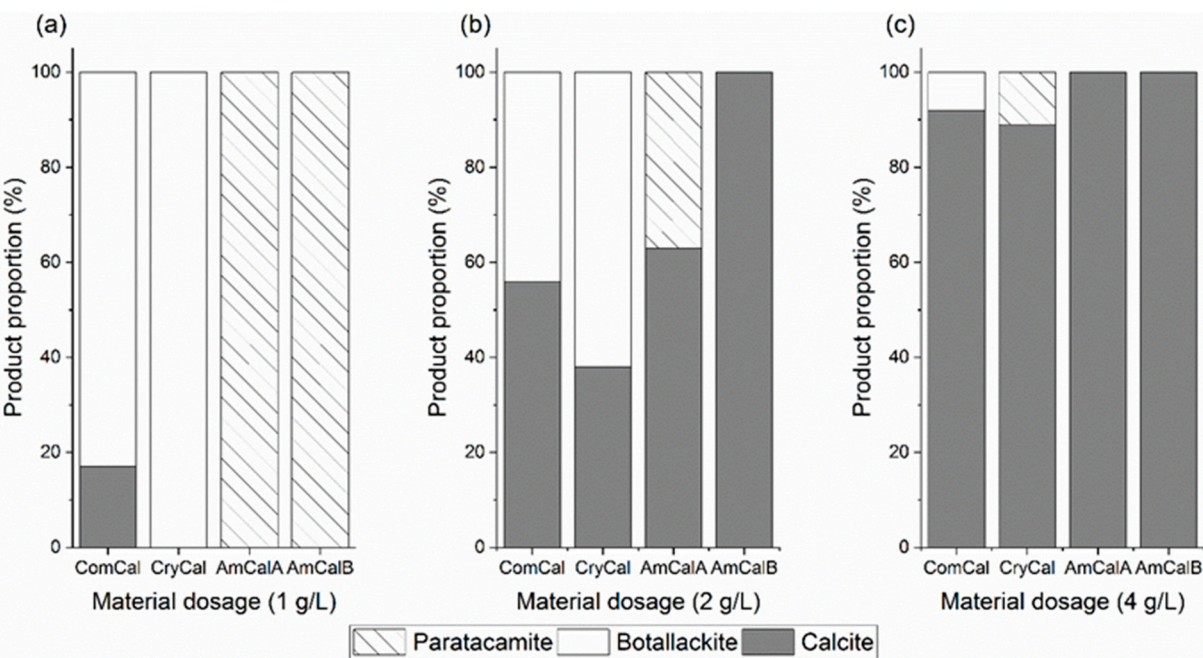

**Figure 5.** The mineral phases proportion of products with different material dosages. (**a**) 1 g/L, (**b**) 2 g/L and (**c**) 4 g/L.

Atacamite, botallackite, clinoatacamite, and paratacamite are the four polymorphs of basic copper chloride ($Cu_2(OH)_3Cl$) [29], with the space group of *Pnma*, *P*$2_1$/m, *P*$2_1$/n, and *R*$\bar{3}$, respectively. As a layered structure, botallackite is less stable than paratacamite, whose structure is a framework. Botallackite will transform into paratacamite once it is left in the mother liquid after precipitation [30–32]. As reported, in a solution with chloride ion ($Cl^-$) concentrations higher than 0.2 mol/L, the initially-formed botallackite recrystallized to paratacamite and atacamite [31]. It is appropriate that the paratacamite precipitated in the ACC system because our ACC contains some halite. In other literature, both botallackite and atacamite precipitated when mechanically activated $CaCO_3$ reacted with $CuCl_2$ solution [16]. However, no atacamite was observed in our experiment. These results implied that the equilibrium and transformation between basic copper chloride are complex and require further investigation.

Accordingly, the reaction mechanism of $CaCO_3$ with copper chloride is mainly precipitation, which the following Equation could describe.

$$2Cu^{2+} + Cl^- + 3OH^- \rightarrow Cu_2(OH)_3Cl \downarrow \tag{7}$$

It is worth noting that calcite was the sole product of ACC with a dosage of 4 g/L and AmCalB at 2 g/L. Because the $Cu^{2+}$ removal efficiencies were nearly 100%, one should wonder where did the removed $Cu^{2+}$ go? Many studies have shown that $Cu^{2+}$ could incorporate into the calcite lattice during the transformation of vaterite into calcite [17–20]. Previously, we also synthesized Cu-bearing calcite using the coprecipitation method [33]. Therefore, our study proposes that the calcite lattice incorporation, i.e., form $(Ca_{1-x}Cu_x)CO_3$ solid solution was the dominant $Cu^{2+}$ removal mechanism at an ACC dosage equal to or higher than 2 g/L as shown in Equation (8).

$$CaCO_3 \cdot nH_2O(ACC) + xCu^{2+} \rightarrow (Ca_{1-x}Cu_x)CO_3(calcite) + (1-x)Ca^{2+} + nH_2O \tag{8}$$

When we combine Equations (5)–(7), we can obtain Equation (9).

$$3CaCO_3 + 3H_2O + 2Cu^{2+} + Cl^- \rightarrow 3Ca^{2+} + 3HCO_3^- + Cu_2(OH)_3Cl \downarrow \tag{9}$$

According to Equation (9), the molar ratio of removed $Cu^{2+}$ and dissolved $Ca^{2+}$ should equal 1.5 if all $Cu^{2+}$ precipitated as $Cu_2(OH)_3Cl$. Meanwhile, the loaded molar ratio of $Cu^{2+}$ and $Ca^{2+}$ were 1.1, 0.6, and 0.3 at a $CaCO_3$ dosage of 1, 2, and 4 g/L, respectively. However, the observed data for ACC was higher than 1.5 (Supplementary Materials Figure S3), indicating less $CaCO_3$ dissolution than the crystalline material (Figure 3d). These data supported the $Cu^{2+}$ removal mechanism via incorporation into calcite, especially at high ACC loading.

### 3.3.3. SEM Observation of Products after $Cu^{2+}$ Removal

Figure 6 shows the morphological features of reaction products of varied $CaCO_3$ with an initial dosage of 1 g/L. ComCal_1 presented a large cubic structure, whereas CryCal_1 showed perfect plate crystals. As discussed previously, botallackite was the sole mineral phase of CryCal_1, whereas ComCal_1 contained some calcite. These observations suggested that the habit of botallackite could be affected by the original habit of crystalline $CaCO_3$. In contrast, single crystals of bipyramid and aggregates of nano prism crystals occurred in AmCalA_1, whereas AmCalB_1 contained nano aggregates only.

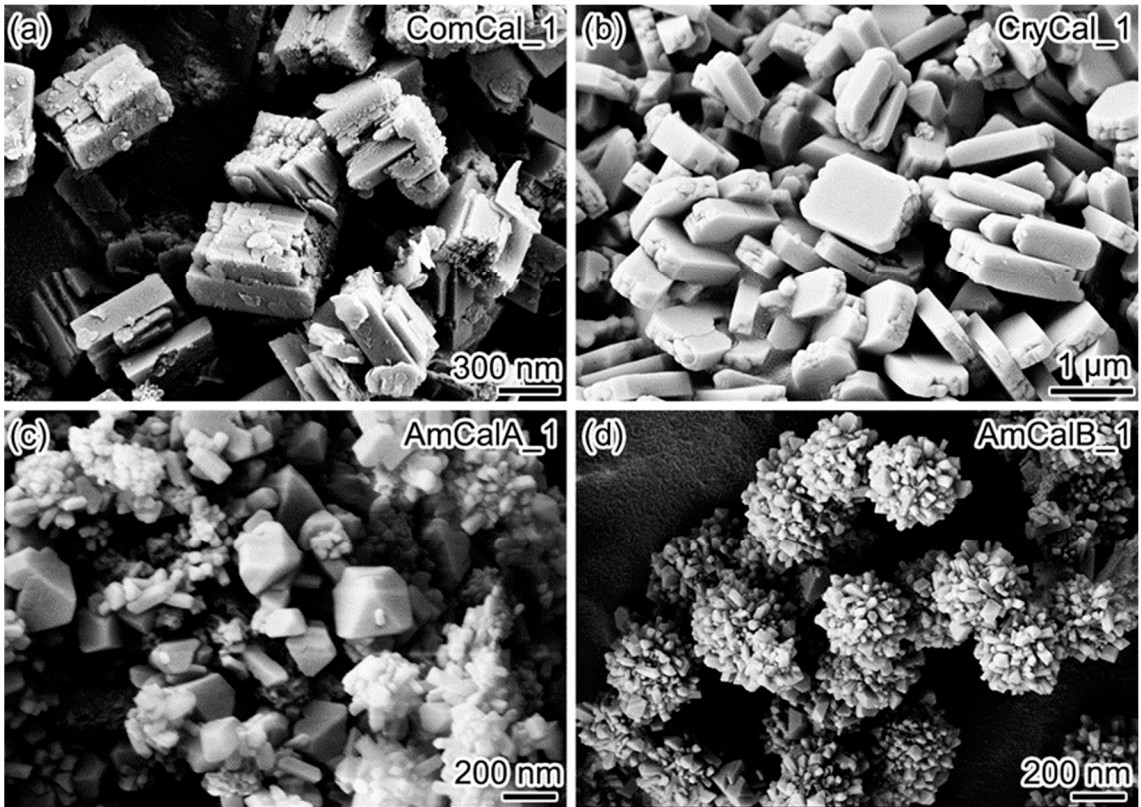

**Figure 6.** The SEM images of ComCal_1 (**a**), CryCal_1 (**b**), AmCalA _1 (**c**), and AmCalB_1 (**d**).

Nevertheless, AmCaA_4 formed dogtooth crystals with a length up to 10 μm (Figure 7a). The corresponding EDS spectrum confirmed the occurrence of Si and Cu, besides Ca (Figure 7b,d). As shown in the XRD results, the sole mineral phase of AmCalA_4 and AmCalB_4 was calcite. These results further supported that calcite could immobilize $Cu^{2+}$ in its lattice via converting from an amorphous precursor.

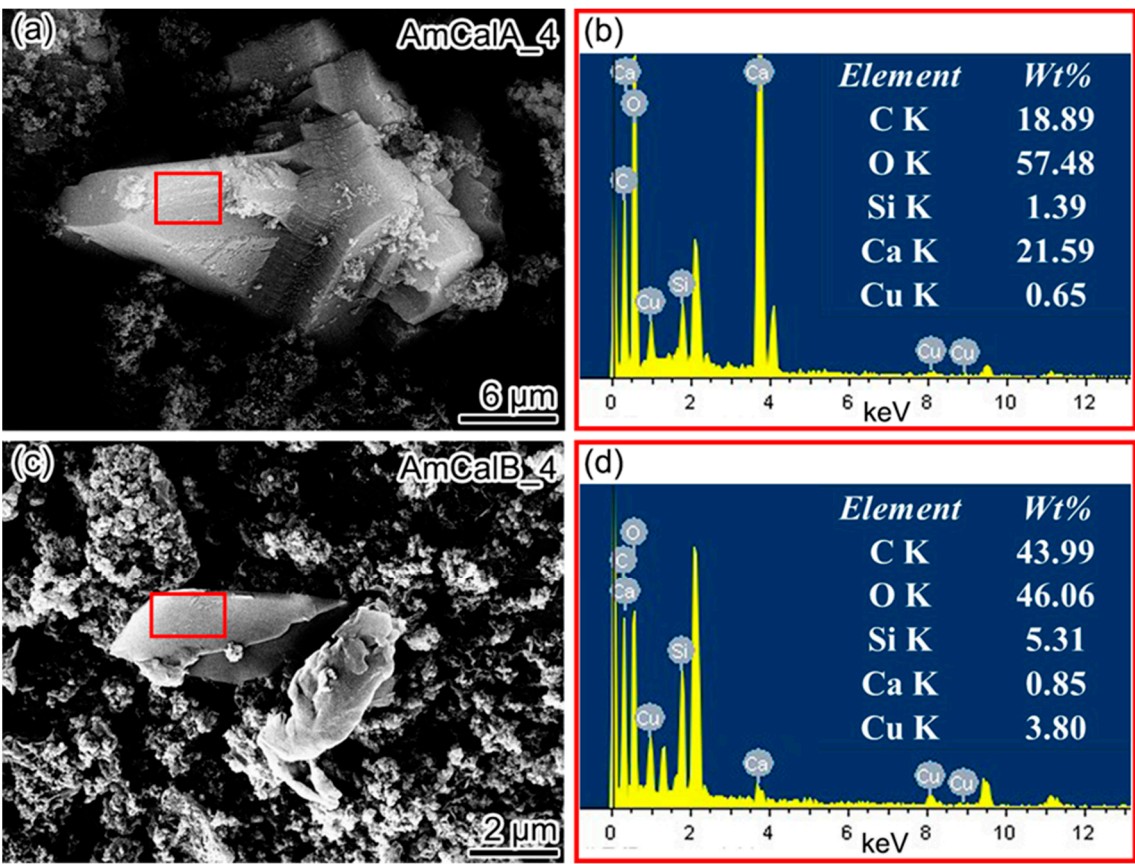

**Figure 7.** The SEM images (**a**,**c**) and EDS (**b**,**d**) analysis of AmCalA_4 and AmCalB_4.

## 4. Conclusions and Implications

Silicon-stabilized amorphous calcium carbonate (ACC) with nano aggregates was successfully synthesized, characterized by XRD and SEM, and used for $Cu^{2+}$ removal. The initial solution pH, initial $Cu^{2+}$ concentration, contacting time, and ACC dosage could impact the $Cu^{2+}$ removal efficiencies. At an ACC dosage of 1 g/L and an initial $Cu^{2+}$ concentration of 700 mg/L at pH 5.0, the removed $Cu^{2+}$ was precipitated as paratacamite, whereas botallackite formed when calcite was used as the starting material. With the increase of ACC dosage, only Cu-bearing calcite was observed as the product, suggesting a novel metal removal mechanism, i.e., metal incorporation into converted calcite via an amorphous precursor. Therefore, our study provided a new understanding of metal scavenges by calcium carbonate materials. Because ACC is present at the early stage of biological calcite and aragonite formation, our results suggested that ACC could be a potential pathway of metal intake by related organisms. Moreover, our silicon-stabilized ACC could be used to recycle asbestos-containing products because calcite and quartz (a polymorph of crystalline $SiO_2$) prevent asbestos decomposing due to the formation of calcium-silicate compounds [34].

**Supplementary Materials:** The following supporting information can be downloaded at: https://www.mdpi.com/article/10.3390/min12030362/s1, Figure S1: Effect of solution pH on the measured concentration of $Cu^{2+}$; Figure S2: Effect of material dosage on the dissolved silicon; Figure S3: Molar ratio of removed Cu and dissolved Ca with different material dosages.

**Author Contributions:** Conceptualization, S.W.; Data curation, Z.L. and S.W.; Funding acquisition, S.W. and F.C.; Investigation, Z.L. and H.Z.; Supervision, S.W. and F.C.; Writing—original draft, Z.L.; Writing—review & editing, S.W. and F.C. All authors have read and agreed to the published version of the manuscript.

**Funding:** This research was funded by the National Natural Science Foundation of China (Grant No. 41877135), Guangdong Basic and Applied Basic Research Foundation (2021A1515011531), and the Science and Technology Planning Project of Guangdong Province, China (Grant No. 2020B1212060055).

**Institutional Review Board Statement:** Not applicable.

**Informed Consent Statement:** Not applicable.

**Data Availability Statement:** Data are available from the authors upon reasonable request.

**Acknowledgments:** This is contribution No. IS-3150 from GIGCAS.

**Conflicts of Interest:** The authors declare no conflict of interest.

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
