# Peer review of "Removal of Aqueous Cu2+ by Amorphous Calcium Carbonate: Efficiency and Mechanism"

_minerals, doi:10.3390/min12030362_

Round 1
Reviewer 1 Report
Review of manuscript: minerals-1631909
This script describes the removal behaviors of amorphous calcium carbonate for copper, and discuss the copper removal mechanism for amorphous calcium carbonate. tIt is an interesting script. There are some points to be clarified in the revised text. The points are described below.
- Abstract: You should add the chemical formula for paratacamite and botallackite to understand easily.
- Abstract, l. 20: “Cu2+ precipitated as paratacamite at an ACC dosage of 1 g/L. At the same time, botallackite was,,,”→” Cu2+ precipitated as paratacamite at an ACC dosage of 1 g/L, while botallackite was,,,”.
- Introduction, l. 52: “such as posnjakite (SO42-), gerhardtite (NO3-), atacamite, and botallackite (Cl-)” You should write chemical formula of each minerals.
- Introduction, l. 68: “the mechanism was proposed” What is the mechanism? Synthesis, adsorption? You should write more detail.
- General: You should compare with amorphous SiO2.
- Results and discussion, l. 148: “which supports the presence of amorphous material.” Why support the amorphous phases? SEM images in Figure 1 shows crystalline phases.
- Results and discussion, l. 156: “contro4lled”→”controlled”
- Results and discussion, 3.1.2, 3.1.3: How is the final pH of the solution after each experiments?
- Results and discussion, 3.3.2: You should add the chemical formula of all minerals in the text.
I recommended publication of this paper, subject to the above points being satisfactorily addressed.
Author Response
Response to reviewer 1
Dear reviewer,
We sincerely appreciate the valuable and helpful comments for improving our manuscript. Accordingly, we have revised the manuscript carefully. Our replies to respective comments are as follows.
- Abstract: You should add the chemical formula for paratacamite and botallackite to understand easily.
Reply: We have revised the manuscript as suggested. The chemical formula of each mineral was provided when the mineral occurred for the first time. We listed the space group of polymorphs with the same formula for better understanding.
- Abstract, l. 20: “Cu2+ precipitated as paratacamite at an ACC dosage of 1 g/L. At the same time, botallackite was,,,”→” Cu2+ precipitated as paratacamite at an ACC dosage of 1 g/L, while botallackite was,,,”.
Reply: We have revised the manuscript as suggested. The new sentence is ‘Cu2+ precipitated as paratacamite (Cu2(OH)3Cl, space group: R ) at an ACC dosage of 1 g/L, while botallackite (Cu2(OH)3Cl, space group: P21/m) was the Cu-bearing product’.
- Introduction, l. 52: “such as posnjakite (SO42-), gerhardtite (NO3-), atacamite, and botallackite (Cl-)” You should write the chemical formula of each mineral.
Reply: We have revised the manuscript as suggested. The new sentence is ‘such as posnjakite (Cu4(SO4)(OH)6·H2O) for SO42-, gerhardtite (Cu2(NO3)(OH)3) for NO3-, atacamite (Cu2(OH)3Cl, space group: Pnma) and botallackite (Cu2(OH)3Cl, space group: P21/m) for Cl-’.
- Introduction, l. 68: “the mechanism was proposed” What is the mechanism? Synthesis, adsorption? You should write more detail.
Reply: We have revised the manuscript as suggested. The new sentence is ‘The Cu2+ removal mechanism by ACC was proposed to be precipitation and incorporation’.
- General: You should compare with amorphous SiO2.
Reply: Thanks a lot for the valuable counsel. It is interesting to compare the metal removal by amorphous SiO2 with ACC, but it is out of our topic more or less. Based on our current data, the Cu2+ removal was ascribed by the reaction between Cu2+ and CaCO3, but not amorphous SiO2. In the future, we would like to investigate the metal removal ability of other amorphous materials, including amorphous SiO2.
- Results and discussion, l. 148: “which supports the presence of amorphous material.” Why support the amorphous phases? SEM images in Figure 1 shows crystalline phases.
Reply: We gratefully appreciate the valuable comment. Figure 1b and Figure 1c show the morphology of our synthesized ACC (AmCalA and AmCalB) similar to the image of SiO2-stabilized ACC in literature, therefore, we said ‘supports the presence of amorphous material’. We have added the reference [28] herein in the revised manuscript. As you can see from Figure 5b in reference [28], the SEM image of ACC is very close to our observation.
The crystalline phases shown in Figure 1 were Figure 1d (CryCal) and Figure 1e (ComCal). Based on the XRD results, CryCal and ComCal are crystalline materials. Their SEM images are shown here for comparison.
[28] Kellermeier, M.; Melero-Garcia, E.; Glaab, F., et al. Stabilization of amorphous calcium carbonate in inorganic silica-rich environments. J Am Chem Soc 2010, 132, 17859-17866, https://doi.org/10.1021/ja106959p.
Figure 5b from Kellermeier, et al. (2010)
- Results and discussion, l. 156: “contro4lled”→”controlled”
Reply: We have corrected the mistake as suggested.
- Results and discussion, 3.1.2, 3.1.3: How is the final pH of the solution after each experiments?
Reply: We are sorry that we did not monitor the final pH after each experiment. We have provided all the pH data in the manuscript we have obtained, which we believed was helpful to understanding the Cu2+ removal mechanism. Probably, we could explain the mechanism better if we obtained all final solution pH as queried. We will keep this suggestion in mind in our future investigation.
- Results and discussion, 3.3.2: You should add the chemical formula of all minerals in the text.
Reply: We have revised the manuscript as suggested. We also added chemical formulas for other minerals in the introduction. When polymorphs existed, a space group was provided.

Reviewer 2 Report
The paper is dealing with the estimation of the potential of the silica stabilized amorphous calcium carbonate (ACC) for the removal of copper ions (Cu2+) from the aqueous media as a function of pH, Cu2+ concentration, contact time and ACC amount. It is a good experimental work which is worth to be published in Minerals after some corrections.
- I recommend (may be for the future work) to use electron paramagnetic/spin resonance techniques to follow the valence states and concentration of the copper complexes either on the surface or as incorporated into the ACC. I think it also can give an answer whether the Copper-ions were embedded into the structure of ACC or crystalline calcite. See, for example, as it was done for calcium phosphates in some recent papers and references therein
Gabbasov, B., Gafurov, M., Starshova, et al .(2019). Conventional, pulsed and high-field electron paramagnetic resonance for studying metal impurities in calcium phosphates of biogenic and synthetic origins. Journal of Magnetism and Magnetic Materials, 470, 109-117.
Fadeeva, I. V., Lazoryak, B. I., Davidova, G. A., et al. (2021). Antibacterial and cell-friendly copper-substituted tricalcium phosphate ceramics for biomedical implant applications. Materials Science and Engineering: C, 129, 112410.
- What about the removal of Cu3+ and Cu+ ions with ACC? I think, this possibility should be also discussed. At least briefly in Introduction. I think ACC is better in this sense than crystalline materials.
Author Response
Response to reviewer 2
Dear reviewer,
We sincerely appreciate the valuable and helpful comments for improving our manuscript. Accordingly, we have revised the manuscript carefully. Our replies to respective comments are as follows.
- I recommend (may be for the future work) to use electron paramagnetic/spin resonance techniques to follow the valence states and concentration of the copper complexes either on the surface or as incorporated into the ACC. I think it also can give an answer whether the Copper-ions were embedded into the structure of ACC or crystalline calcite. See, for example, as it was done for calcium phosphates in some recent papers and references therein
Gabbasov, B., Gafurov, M., Starshova, et al .(2019). Conventional, pulsed and high-field electron paramagnetic resonance for studying metal impurities in calcium phosphates of biogenic and synthetic origins. Journal of Magnetism and Magnetic Materials, 470, 109-117.
Fadeeva, I. V., Lazoryak, B. I., Davidova, G. A., et al. (2021). Antibacterial and cell-friendly copper-substituted tricalcium phosphate ceramics for biomedical implant applications. Materials Science and Engineering: C, 129, 112410.
Reply: Thanks a lot for the valuable suggestion. Unfortunately, we are currently unable to characterize our material using electron paramagnetic/spin resonance techniques. We will try it in our future investigation.
- What about the removal of Cu3+ and Cu+ ions with ACC? I think, this possibility should be also discussed. At least briefly in Introduction. I think ACC is better in this sense than crystalline materials.
Reply: As suggested, we added a new sentence in the second paragraph to show the abundance of Cu2+ in nature. The new sentence is ‘Cu exists in the forms of Cu+, Cu2+, and Cu3+ ions, among which Cu2+ is the most stable and widespread in the environment [5,6]’. However, we did not compare the removal of Cu3+ and Cu+ by ACC since it is out of our topic.
Reviewer 3 Report
The paper reports on the Cu removal ability of amorphous calcium carbonate vs that of the parent crystallized compound. It is informative and consistently written. However the products characterization by means of XRD is not convincing considering the quantitative analysis.
The RIR method is based upon scaling all diffraction data to the diffraction of standard reference materials. The assumption is that all the factors, except concentration, of the analyte are ratioed and reduced to a constant. It is known that synthesized materials (such as those investigated in the paper) often contain trace impurities, anion/cation substitution in the lattice, vacancies, strain that will result in preferred orientation therefore alter peak intensity.
These factors (likely to influence peak intensities and peak profiles) will change the I/c value which can be significantly different from one calculated from a perfect crystalline pure material. In this respect the authors should carefuly comment the limits of the semi-quantitative analysis performed by XRD. The manuscript needs therefore a revision.
Author Response
Response to reviewer 3
Dear reviewer,
We sincerely appreciate the valuable and helpful comments for improving our manuscript. Accordingly, we have revised the manuscript carefully. Our replies to respective comments are as follows.
The paper reports on the Cu removal ability of amorphous calcium carbonate vs that of the parent crystallized compound. It is informative and consistently written. However the products characterization by means of XRD is not convincing considering the quantitative analysis.
The RIR method is based upon scaling all diffraction data to the diffraction of standard reference materials. The assumption is that all the factors, except concentration, of the analyte are ratioed and reduced to a constant. It is known that synthesized materials (such as those investigated in the paper) often contain trace impurities, anion/cation substitution in the lattice, vacancies, strain that will result in preferred orientation therefore alter peak intensity.
These factors (likely to influence peak intensities and peak profiles) will change the I/c value which can be significantly different from one calculated from a perfect crystalline pure material. In this respect the authors should carefully comment the limits of the semi-quantitative analysis performed by XRD. The manuscript needs therefore a revision.
Reply: We gratefully appreciate the rigorous comment. It is essential to emphasize the limits of the semi-quantitative analysis performed by XRD. In the revised manuscript (Section 3.3.2), we have pointed out the limits of our semi-quantitative analysis. The newly added sentence is ‘It is worth noting that the above semi-quantitative results of XRD are not accurate and are only a general reference because materials containing trace impurities will alter peak intensity’ which can be found at the end of the second paragraph in Page 8.
Reviewer 4 Report
The paper is a honest and linear study on the importance of CaCO3 in the removal of copper form contaminated water. The paper is well expalined and all the conclusions are supported by the results. The only observation is that the work was carried out on an artificially contaminated water. Real contaminated waters can present unknown difficulties.
A few remarks:
Line 85: which was the particle-size of ACC wjen it was mixed with the contaminated water ?
Line 129: the limits of R are 0 and infinity ? The definition of R should be a depletion rate (...the concentration is decreased by 400 %...., for example). The Removal efficiency should range between o and 100 as in the following figures is reported.
The authors could add this reference as an example that calcium carbonate can also be used in other fields of the environmental protection, decomposition of asbestos in this case.
Influence of calcium carbonate on the decomposition of asbestos contained in end-of-life products. Belardi G. and Piga L. Thermochimica Acta, 2013, 573, pp 220-228. https://doi.org/10.1016/j.tca.2013.08.019
Author Response
Response to reviewer 4
Dear reviewer,
We sincerely appreciate the valuable and helpful comments for improving our manuscript. Accordingly, we have revised the manuscript carefully. Our replies to respective comments are as follows.
- The only observation is that the work was carried out on an artificially contaminated water. Real contaminated waters can present unknown difficulties.
Rely: We gratefully appreciate the rigorous comment. Reference [16] has demonstrated that different anions could result in different precipitation. We agree with the reviewer that a real wastewater might have a different result. Therefore, we would like to treat real contaminated waters using our ACC in the future.
- Line 85: which was the particle-size of ACC when it was mixed with the contaminated water ?
Reply: We have revised the manuscript as suggested and provided the particle-size information of ACC in Section 2.2. The revised sentence is ‘ACC passed through a 0.25 mm sieve was mixed with copper chloride solution’.
- Line 129: the limits of R are 0 and infinity? The definition of R should be a depletion rate (...the concentration is decreased by 400 %...., for example). The Removal efficiency should range between o and 100 as in the following figures is reported.
Reply: We have corrected the manuscript as suggested in Section 2.4. The revised sentence is ‘The Cu2+ removal efficiency (R∈[0, 100], %)…’.
- The authors could add this reference as an example that calcium carbonate can also be used in other fields of the environmental protection, decomposition of asbestos in this case.
Influence of calcium carbonate on the decomposition of asbestos contained in end-of-life products. Belardi G. and Piga L. Thermochimica Acta, 2013, 573, pp 220-228. https://doi.org/10.1016/j.tca.2013.08.019
Reply: We have revised the manuscript as suggested at the end of Section 4. The newly added text is ‘Moreover, our silicon stabilized ACC could be used to recycle asbestos-containing products because calcite and quartz (a polymorph of crystalline SiO2) prevent asbestos de-composing due to the formation of calcium-silicate compounds [34].’
Round 2
Reviewer 3 Report
it looks that in the revised version the suggestions and amendments of the reviewer have been taken care of. The paper is now worth of publication